# Congenital Aneuploidy in Klinefelter Syndrome with B-Cell Acute Lymphoblastic Leukemia Might Be Associated with Chromosomal Instability and Reduced Telomere Length

**DOI:** 10.3390/cancers14092316

**Published:** 2022-05-06

**Authors:** Eigil Kjeldsen

**Affiliations:** Cancercytogenetics Section, Department of Hematology, Aarhus University Hospital, DK-8200 Aarhus, Denmark; eigil.kjeldsen@clin.au.dk; Tel.: +45-40146175

**Keywords:** Klinefelter syndrome, chromosomal instability, telomere length, hematological malignancy, iQ-FISH

## Abstract

**Simple Summary:**

Klinefelter syndrome (KS) is a rare congenital aneuploidy characterized by inherited gain of one X chromosome (XXY). KS is associated with higher susceptibility to the development of cancer. Somatic acquired chromosomal aberrations and chromosomal instability are hallmarks of cancer and leukemia but little is known about the cellular mechanisms involved. The conducted research aimed to identify genomic mechanisms involved in chromosomal evolution mechanisms important for leukemic development. In the leukemic blasts of a patient with KS and B-cell acute lymphoblastic leukemia (B-ALL), we identified additional acquired chromosomal aberration and a significant reduction in the length of the chromosomal ends, i.e., telomeres. A literature review of KS patients with B-ALL revealed that the majority of these patients had acquired two or more additional chromosomal aberrations at B-ALL diagnosis. These data indicate that enhanced reduction in telomere length might be associated with chromosomal instability and may serve as a future target for therapy or prevention.

**Abstract:**

Rare congenital aneuploid conditions such as trisomy 13, trisomy 18, trisomy 21 and Klinefelter syndrome (KS, 47,XXY) are associated with higher susceptibility to developing cancer compared with euploid genomes. Aneuploidy frequently co-exists with chromosomal instability, which can be viewed as a “vicious cycle” where aneuploidy potentiates chromosomal instability, leading to further karyotype diversity, and in turn, paving the adaptive evolution of cancer. However, the relationship between congenital aneuploidy *per se* and tumor initiation and/or progression is not well understood. We used G-banding analysis, array comparative genomic hybridization analysis and quantitative fluorescence in situ hybridization for telomere length analysis to characterize the leukemic blasts of a three-year-old boy with KS and B-cell acute lymphoblastic leukemia (B-ALL), to gain insight into genomic evolution mechanisms in congenital aneuploidy and leukemic development. We found chromosomal instability and a significant reduction in telomere length in leukemic blasts when compared with the non-leukemic aneuploid cells. Reviewing published cases with KS and B-ALL revealed 20 additional cases with B-ALL diagnostic cytogenetics. Including our present case, 67.7% (14/21) had acquired two or more additional chromosomal aberrations at B-ALL diagnosis. The presented data indicate that congenital aneuploidy in B-ALL might be associated with chromosomal instability, which may be fueled by enhanced telomere attrition.

## 1. Introduction

Klinefelter syndrome (KS) is a congenital aneuploid syndrome characterized by an additional X chromosome with an incidence of 150 in 100,000 live-born males [1]. The 47,XXY karyotype represents the most common chromosomal pattern in KS (80%), while the remaining patients display mosaic patterns (20%) [1,2]. Patients with KS are most often diagnosed after puberty because of the subtlety of clinical manifestations, especially in children or young patients [2].

Patients with congenital aneuploidy have increased rates of chromosomal instability (CIN). CIN is generally defined as the propensity of cells to give rise to progeny with an altered chromosome number or structure when compared with parental cells. Previous studies showed that cells derived from patients with trisomy 13 (Patau syndrome), trisomy 18 (Edwards syndrome), trisomy 21 (Down’s syndrome) or monosomy X (Turner syndrome) exhibit a significantly higher level of acquired sporadic non-chromosome-specific losses or gains of whole chromosomes when compared with euploid controls [3,4]. Studies using artificially generated trisomies and tetrasomies in human cells revealed that the addition of even a single chromosome in human cells promotes genomic instability by increasing DNA damage and sensitivity to replication, and it elevates the occurrence of chromosomal abnormalities by chromosome segregation errors [5]. Together, these observations support a previously proposed hypothesis that aneuploidy catalyzes chromosomal instability [6,7,8], although the mechanism(s) for such an autocatalytic process remains unknown.

Aneuploidy is frequent in human cancers and is often associated with CIN, which correlates with metastasis, resistance to drugs and disease progression [9,10,11,12,13]. Congenital aneuploidies are associated with an increased risk of certain malignancies [14]. Down’s syndrome patients have a high risk of acute myeloid leukemia, particularly the megakaryoblastic subtype [15,16], and children with Edwards syndrome (trisomy 18) are at increased risk of developing Wilms’ tumor and hepatoblastoma [17]. In the group of sex chromosome aneuploidies including monosomy X (Turner syndrome), triple X, Y polysomy and XXY (KS), some of these have an elevated risk of several malignancies. Female Turner syndrome patients have an increased risk of gonadoblastoma and brain tumors [18], while men with Y polysomy have rates of cancer incidence equivalent to the general population [19]. Men with KS have elevated risks of several solid tumors such as lung cancer, breast cancer and non-Hodgkin’s lymphoma [17,18], while data on the risk of leukemic malignancies in KS are conflicting. Some reports suggest an increased risk of leukemia development in males with KS, while others have deemed it only a chance association [20]. A recent study on B-cell acute lymphoblastic leukemia (B-ALL) reported that males with KS are not at increased risk of developing B-ALL and even suggested a protective effect of KS on the development of B-ALL [21]. Although congenital aneuploid cells are linked with increased rates of cancer, exactly how aneuploidy affects the cells and how it contributes to tumorigenesis are only partly understood.

Here, we describe a three-year-old boy whose diagnosis of KS was established during the initial cytogenetic workup for B-ALL. We characterized his leukemic blasts by G-banding, FISH and array comparative genomic hybridization (aCGH). In addition, we performed a search of the literature and identified 21 additional KS cases with B-ALL and a karyotype at diagnosis of B-ALL. A review of their cytogenetic workup was performed to examine whether these patients exhibited CIN. Together, the cytogenetic data from our KS patient and the review of the literature, we suggest that congenital aneuploidy with a constitutional extra X chromosome is associated with CIN, which in turn, seems to be associated with short telomeres.

## 2. Materials and Methods

### 2.1. G-Banding Analysis

G-banded chromosomes were prepared after short-term unstimulated culturing of bone marrow cells at diagnosis and follow-up, and after PHA-stimulated peripheral blood cells at remission, according to standard procedures [22]. Karyotyping was done using a charged coupled device (CCD)-based imaging system IKAROS (Metasystems, Altlussheim, Germany). Karyotypes were described according to ISCN2020 [23]. Residual cell suspensions used for conventional cytogenetic examination were stored at −20 °C. Unfortunately, no material remained after performing the additional comprehensive analyses as described below.

### 2.2. Fluorescence In Situ Hybridization (FISH) Analysis

FISH analyses were done on the same bone marrow suspension as previously used for G-banding analyses from the time of diagnosis. Human multicolor FISH was carried out according to the manufacturer’s instructions using the 24XCyte kit consisting of 24 different chromosome painting probes (Metasystems). Each of the XCyte probes was labeled with one of five fluorochromes or a unique combination thereof (combinatorial labeling). Metaphases were counterstained with 4′,6-diamidino-2-phenylindole (DAPI)/antifade solution. Images were captured using an automated Zeiss Axio Imager-Z2 equipped with a CCD camera (CoolCube1) and appropriate filters, along with Isis software (MetaSystems). Karyotyping was done using the 24-color mFISH upgrade package ISIS.

Locus-specific and whole-chromosome painting analyses were carried out with the following directly labeled probes according to the manufacturer’s instruction: (1) pre-B-ALL FISH panel including the following dual-color probes: ABL1 break apart (ZytoVision GmbH, Bremerhaven, Germany), ABL2 break apart (MetaSystems), TCF3 (CytoCell Ltd., Oxford Gene Technology, Cambridge, UK), ETV6/RUNX1 dual fusion (Abbott Molecular Inc., Des Plaines, IL, USA), KMT2A break apart (Abbott Molecular Inc.), PDGFRB break apart (Kreatech Biotechnology B.V., Amsterdam, The Netherlands) and CSFR1 (ZytoVision GmbH), (2) dual-color split apart CCND2 (CytoCell Ltd., Oxford Gene Technology), (3) subtelomeric probes for: 7pter, 7qter and 12pter (all from Kreatech Biotechnology B.V.), (4) TWIST1 and SKAP2 custom-made BAC-based probes (Empire Genomics, New York City, NY, USA) and (5) whole-chromosome painting probes for chromosomes 7 and 12 (Kreatech Biotechnology B.V.). Metaphase chromosomes and interphase nuclei were counterstained with DAPI/antifade solution.

### 2.3. Quantification of Telomere Content

The telomere content was measured in interphase nuclei (iQ-FISH) and metaphases (mQ-FISH) by T/C (telomere/centromere ratio) FISH analysis on the stored bone marrow suspension, as previously used for G-banding and FISH analyses at the time of diagnosis. T/C FISH analysis has the advantage of co-hybridization with the pan-telomeric (G_3_TA_2_) Cy-3 labeled PNA probe, and as an internal reference, a chromosome 2 centromere-specific FITC-labeled PNA probe (Agilent Technologies, Santa Clara, CA, USA) to eliminate variations due to uneven hybridization efficiency.

iQ-FISH was performed using the pan-telomeric (G_3_TA_2_) Cy-3-labeled PNA probe, and as an internal reference, a chromosome 2 centromere-specific FITC-labeled PNA probe (Agilent Technologies), as described in [24,25,26]. Briefly, slide scanning, intensity measurement and quantification were performed using the automated fluorescence-based microscopic scanning system Metafer4 (MetaSystems, Altlussheim, Germany). The Metafer4 consists of a fluorescence microscope (Axio.Imager.Z2, Zeiss, Jena, Germany) with individual filter sets for DAPI, FITC and Cy3, equipped with a CCD camera (CoolCube1, MetaSystems) and linked to the ISIS and MetaCyte software (MetaSystems). The MetaCyte software allows for automated capture at fixed exposure times for each color channel as well as image- and cell-processing steps. The FITC and Cy3 signal intensities in single nuclei were measured using the 63× objective in nine focus planes, and the ratio between both intensities (Cy3/FITC) multiplied by 100 was automatically calculated and recorded in fluorescence ratio units (FRUs), representing the telomere content. A minimum of 250 nuclei were scanned and recorded. The system allows for relocation and visual inspection of scanned nuclei to evaluate and exclude possible artifacts. To compare the telomere content in interphase nuclei at diagnosis harboring only the extra X chromosome, or in interphase nuclei with additional chromosomal aberrations, the scanned slide used for telomere measurement was stripped and re-hybridized with a cocktail of probes for 7qter, 12pter and SKAP2. The Metafer4 system was used to relocate previously recorded nuclei included in the telomere measurement, and images of all relocated nuclei were captured on the Metafer4 system using the ISIS software (MetaSystems). Nuclei were manually scored with respect to the hybridization pattern they contained (+X: 4R2G; +X, +Add: ≥ 6R2G), and the corresponding FRU value was recorded.

mQ-FISH allows for quantification of the telomere content in individual metaphases and individual chromosome arms with respect to their genomic complement. This was done using the pan-telomeric (G_3_TA_2_), Cy-3 labeled PNA probe, and as an internal reference, a chromosome 2 centromere-specific FITC-labeled PNA probe (Agilent Technologies) as described for iQ-FISH. After the hybridization and washing steps, automated slide scanning was performed using the automated fluorescence-based microscopy scanning system Metafer4 (MetaSystems), in which the MetaCyte software allowed for automated metaphase-finding and capture of miniature metaphase images. This allowed for relocation and visual inspection of scanned metaphases, to evaluate and exclude possible artifacts. Then, approved metaphases were captured automatically at fixed exposure times for each color channel using the linked ISIS and MetaCyte software (MetaSystems), and images were stored in the TIFF format. A minimum of 50 metaphases were initially recorded. To compare the telomere content in metaphases harboring only the extra X chromosome +X with metaphases with additional chromosomal aberrations +X, +Add, the scanned slide used for telomere capture was stripped and re-hybridized with a cocktail of probes representing 7qter, 12pter and SKAP2. The Metafer4 system was used to relocate previously recorded metaphases included in the telomere capture, and new corresponding images after re-hybridization were captured and stored.

The telomere contents in whole metaphases were quantitated on stored images of metaphases after T/C hybridization. First, the stored RGB images were split into grayscale images corresponding to their red, green and blue color channels. Corresponding red and green grayscale images were imported to ImageJ freeware version 1.46r in the TIFF format for measuring/determining red and green signal intensities after background subtraction using quantitative tools in Image J (ImageJ. Available online: https://imagej.nih.gov/ij/download.html, accessed on 16 January 2020). Then, 32 metaphases were quantitated and grouped according to their re-hybridization pattern, i.e., +X and +X, +Add groups. Statistical analysis of grouped T/C FRU values was carried out in GraphPad Prism.

The images used for telomere quantification of whole metaphases were also used to quantitate the telomere content in individual chromosome arms in the recorded greyscale metaphases (32 red and 32 green) using Image J. We examined individual chromosome arms from normal chromosomes, including 7p, 7q, 12p and 12q, and those involved in the structural rearrangements between chromosomes 7 and 12, including der(7p), der(7q), der(12p) and der(12q), and as controls, the telomeres of 1p, 1q, 2p and 2q. This can be done because individual chromosomes can easily be identified by their unique normal and aberrant hybridization pattern after re-hybridization, and for chromosomes 1 and 2, by their morphology and original centromere 2 hybridization in the +X and +X, +Add metaphases. The signal intensities for each telomere set were background-corrected and normalized. Statistical analysis of grouped T/C FRU values was carried out in GraphPad Prism.

### 2.4. Probestripping Followed by Re-Hybridization FISH Analysis

To remove hybridized probes from the previously hybridized slides, the slides were first washed in 2X SSC for 2 min at room temperature to remove immersion oil, coverslips and antifade glycerol [26]. Secondly, two incubations in denaturation solution (1× Tth buffer supplemented with 8.7% glycerol) for 3 min each at 93 °C, with a brief wash in 2× SSC at room temperature between the denaturation steps, were carried out to remove the probes, which was highly efficient as evaluated by FISH microscopy. After probe removal and denaturation, the slides were dehydrated in an ice-cold ethanol series: 70%, 90% and 100% for 3 min each, and air-dried. Re-hybridization with relevant FISH probes was then conducted according to the manufacturer’s instructions.

### 2.5. Oligo-Based Array Comparative Genomic Hybridization (aCGH) Analysis

aCGH analysis was performed using CytoChip 4× 180K v2.0 (Agilent Technologies Denmark ApS, Glostrup, Denmark). The analysis was conducted according to the manufacturer’s instructions using 0.5 µg patient DNA from bone marrow cells at diagnosis and peripheral blood cells at remission, as described in [27]. After hybridization and washing, the oligo array was scanned at 2.5 µm with a GenePix 4400A microarray scanner. Initial analysis and normalization were then carried out with Agilent CytoGenomics version 3.0.6.6 (Agilent Technologies). For analysis and visualization, normalized log2 probe signal values were imported into Nexus Copy Number software version 7.5 (BioDiscovery, El Segundo, CA, USA) and segmented using the FASST2 segmentation algorithm with a minimum of three probes/segment. Regions of gains or losses contained within copy number variable regions (CNVs) were discarded. The human reference genome was NCBI build 37 (hg19). Bioinformatics analysis was performed by querying the UCSC database (UCSC database. Availabel online: https://genome.ucsc.edu, accessed on 15 March 2021).

### 2.6. Statistical Analysis

Statistical analyses were performed using GraphPad Prism software version 8.0 (GraphPad Prism Software Inc., La Jolla, CA, USA). Unpaired t-testing was performed to evaluate the significance of various telomere contents in metaphases and individual chromosome arms between +X and +X, +Add cells. Values of *p* < 0.05 were considered statistically significant.

### 2.7. Literature Search

The literature survey was performed using the online portal for the National Library of Medicine (PubMed.gov. Available online: https://pubmed.ncbi.nlm.nih.gov, accessed on 15 March 2022). The search terms were “Klinefelter syndrome & leukemia”, “Klinefelter & leukemia”, “Klinefelter syndrome and cancer” and “Klinefelter and cancer”. Some cases were also found from cases referenced in retrieved publications.

### 2.8. Clinical Case Description

A male was born at the gestational age of 40 weeks as the first-born child of a healthy Caucasian couple. The proband’s birth weight was 3464 g (normal range, 95%: 2902–3770 g), length was 50 cm (normal range, 95%: 47.4 to 5.4 cm) and head circumference was 36 cm (normal range, 95%: 33.4 to 35.6 cm). The APGAR scores were 10/10/10 for 1/5/10 min, respectively. No dysmorphic features were noted and development in the following years adhered to normal standards. At the age of three years and three months, the patient had an episode of bilateral retention of testes and was admitted to a urologist for a workup where it was noticed that both testes could be descended and that the right testis was smaller than the left one. It was also noted that both testes were palpable at birth and the two-year physical examination, without additional workup.

At the age of three years and eight months, the child was diagnosed with pre-B ALL after three weeks with a history of pain in the right lower leg interpreted as arthritis and treated with oral anti-inflammatory drugs (paracetamol and naproxen). He also experienced flu-like symptoms with a high fever, coughing, night sweats, fatigue and slight abdominal pain. The coughing and fatigue had been present for a couple of months before admission to the pediatric hematologic ward. A physical examination at admission revealed no dysmorphic signs and that he was rather well with only a slight cough and no signs of pneumonia, only a slight fever and a few very small petechiae on his right lower leg.

The patient’s hematological parameters at admission were white blood cells (WBC) at 5.2 × 10^9^/L, consisting of 7% neutrophils, 85% lymphocytes, 7.5% monocytes and 0.6% basophils. The red blood cell (RBC) count was 3.6 × 10^12^/L (normal values 3.3–6.0), hemoglobin level 6.1 mmol/L (normal values 6.5–8.9) and platelet count 11 × 10^9^/L (normal values 165–435). The plasma lactate dehydrogenase (LDH) value was 273 U/L (normal values 155–450), plasma alanine transaminase (ALAT) value was 103 U/L (normal values 5–45) and plasma alkaline phosphatases value was 205 (normal values 130–385). The C-reactive protein (CRP) value was 126.9 mg/L (normal value below 8.0 mg/L) and plasma ferritin value was 370 µg/L (normal values 10–118). Immunophenotyping by flow cytometry (FCM) revealed a leukemic cell population in the bone marrow aspirate and peripheral blood with the following immunophenotype: CD19+, CD45–, CD34+, TdT+, CD10+, CD22+, CD20–, CD38–, CD58–, which was consistent with pre-B-ALL (FAB classification). The morphology was consistent with this interpretation and showed 82% leukemic cells.

The patient was treated according to the NOPHO IR-high risk ALL chemotherapy protocol (ALLTogether protocol) according to the cytogenetic findings (see below), where treatment was initiated the following day after admission. The patient obtained complete remission within 30 days after diagnosis, as determined by FCM, morphology and cytogenetics. The patient was in continuous complete remission at 27 months of follow-up.

## 3. Results

The standard diagnostic cytogenetic workup of patients with a pre-B ALL diagnosis includes initial chromosome analysis by G-banding of aspirated bone marrow cells after unstimulated 24-h culturing, an array of targeted locus-specific FISH analyses and oligo-based DNA microarray analysis.

G-banding revealed an unbalanced karyotype, 47,XY,+X,del(7)(p13)[4]/47,idem,+10,– 15[8]/47,XY,+X[13], indicating hyperdiploidy with 47 chromosomes and three related subclones (Figure 1A).

The common aberration between the three subclones was an additional X chromosome, and two of the subclones harbored an apparent terminal deletion on the short arm of chromosome 7, del(7)(p13), while another subclone had additional trisomy 10 and monosomy 15. Targeted locus-specific interphase nuclei FISH analyses with a B-ALL iFISH panel (see Materials and Methods) showed that in 91% of interphase nuclei, the ETV6 locus harbored a mono-allelic deletion, while other targeted FISH analyses were normal. Next, 24-color karyotyping confirmed the findings by G-banding but unexpectedly revealed a cryptic, apparently unbalanced insertion of chromosome 7 material to the short arm of one chromosome 12 (Figure 1B). It is well-known that whole-chromosome painting has a higher resolution than 24-color karyotyping and that combinatorial artifacts may exist. We, therefore, performed dual-color whole-chromosome painting with chromosome 7 and 12 painting probes, which first of all, confirmed the inserted chromosome 7 material on der(12) near its centromere, and further revealed that it was a double insertion (Figure 1C). In addition, we found that chromosome 12 material was also present on the terminal part of the derivative chromosome 7, indicating an unbalanced translocation between chromosomes 7 and 12 as well (Figure 1C).

aCGH analysis confirmed the numerical aberrations found by G-banding and FISH analyses, including the additional X chromosome, trisomy 10 and monosomy 15, and also revealed 18 additional sub-chromosomal aberrations (Figure 2, Table 1), including: (1) an approximately 0.58-Mb mono-allelic deletion on 5q33.3 involving the EBF1 gene; (2) in the 7p12.1p22.22 region, several deletions and gains were observed, equivalent to chromothripsis involving most of the short arm of chromosome 7, resulting in mono-allelic deletion of IKZF1 and TWIST1 genes and high-copy-number gain of the SKAP2 gene among deletions or duplications of other genes at 7p; (3) an approximately 11.28-Mb duplication of the chromosomal region 12p13.33 to 12p13.2 including the CCND2 gene; (4) an approximately 16.47-Mb mono-allelic deletion of the chromosomal region 12p31.2 to 12p11.23 including the ETV6 gene.

To further characterize the chromosomal aberrations, additional locus-specific FISH studies were performed. First, we wanted to examine the subtelomeric regions of the der(7p) and der(12p) chromosomes using the subtelomeric probes 7pter and 12pter in a dual-color experiment (Figure 3A). As expected, the 12pter probe was localized at the terminal end of the short arms of the normal chromosome 12 and the derivative der(12p), but it was also localized at the telomeric end of the chromothriptic chromosome 7, der(7p), thereby confirming the results from the dual-color whole-chromosome experiment, as shown in Figure 1C.

The 7pter probe was absent from the chromothriptic chromosome der(7p), but as expected, present on the normal chromosome 7. We next wanted to examine which regions of chromosome 7 were involved in the translocation to der(12). For this purpose, we stripped the microscopy slide used for the 7pter/12pter experiment and re-hybridized the slide with the locus-specific probes including the TWIST1 (7p21.1) and SKAP2 (7p15.2) genes (Figure 3A). We found that these genes, as expected, were present on the normal chromosome 7 and were absent on the chromothriptic del(7p) chromosome. Surprisingly, we found that the SKAP2 gene was localized on the short arm of der(12), and was highly amplified, which confirmed the aCGH findings. To further characterize the cytogenetic aberrations involving chromosome 12, we used the dual-color split apart probe CCND2 (12p13.32) in another hybridization experiment. We found three hybridization signals of CCND2, one of which was located on the chromothriptic der(7p) chromosome, another on der(12p) and the last on the normal chromosome 12 (Figure 3B).

After induction chemotherapy, peripheral blood samples were drawn approximately one month later for remission evaluation. The FCM and morphology independently showed complete remission, as did relevant biochemistry analyses. We performed G-banding on unstimulated cultures from the peripheral blood and found that the previously identified additional X chromosome persisted, while the previously identified trisomy 10, monosomy 15, der(7)t(7;12) and der(12)ins(7;12) had all disappeared, including the ETV6 deletion. These findings indicate either partial cytogenetic remission or a congenital additional X chromosome. To discriminate between these possibilities, we performed a PHA stimulated culture of peripheral blood, which confirmed the additional X chromosome with no other chromosomal aberrations in metaphases. aCGH analysis on isolated white blood cells from the peripheral blood confirmed the persisting additional X chromosome, along with the absence of all previously identified genomic aberrations (data not shown).

Considering our cytogenetic findings together, they were consistent with the congenital condition of an additional extra X chromosome, which is associated with KS. Interphase nuclei FISH with probes for CenX and Yq12 using the PHA stimulated culture revealed a mosaic condition with 98% XXY cells and 2% XY cells. We concluded that the patient had mosaic KS, although the pediatricians found no clinical signs of KS except for the episode with retracted testicles (see clinical description).

The apparent, rather simple rearrangement observed by G-banding turned out to be much more complex, as evidenced by aCGH and FISH findings. A schematic representation summarizing the complex findings is shown in Figure 4. A revised karyotype summarized the main experimental findings as follows: 47,XXYc [6]/47,idem,der(7)t(7;12)(p12.1;p13.32),der(12)ins(12;7)(p11.23;p?p?) or ins(12;7)(p13.31;p?p?) [9]/47,sdl,+10,-15.

It has been suggested that a short telomere length is involved in chromosomal instability, leading to chromosomal aberrations [28,29]. To examine the telomere length in our KS patient, we applied the T/C iQFISH method combined with probe stripping and re-hybridization with probes for SKAP2, 7qter and 12pter, to discriminate between +X and +X, +Add nuclei (Figure 5). We analyzed using T/C FISH a total of 258 nuclei, in which 33 had +X and 225 had +X, +Add after re-hybridization. We found that the telomere content in interphase nuclei with +X, +Add was significantly lower (about 3.3 times, *p* < 0.0001) when compared with cells with isolated +X.

To examine whether the significantly reduced telomere content in nuclei with +X, +Add compared with nuclei with isolated +X also is displayed in dividing cells, we applied the T/C mQ-FISH method to metaphases, combined with probe stripping and re-hybridization with probes for SKAP2, 7qter and 12pter, to discriminate between +X and +X, +Add nuclei (Figure 6). We analyzed using T/C FISH a total of 32 metaphases, in which 7 had +X and 25 had +X, +Add after re-hybridization. The lower telomere content was confirmed in metaphase cells, although the magnitude of difference was slightly less (Figure 6B).

It has previously been shown that the telomere length correlates with the size of the associated chromosome arm in normal individuals [30]. We, therefore, wanted to examine whether the lower telomere content in +X, +Add metaphases could be ascribed to specific chromosomes or chromosome arms, or if it was part of a more generalized phenomenon. To examine the telomere length in individual chromosome arms, we re-examined 32 images of the metaphases used for analysis of the telomeric content in metaphases with +X alone and in metaphases with +X, +Add (Figure 6A). First, we examined the relative telomere length in chromosome arms 7p, 7q, 12p and 12q, which are involved in the additional chromosomal aberrations, as described previously, and as controls, in 1p, 1q, 2p and 2q in metaphases with +X as the sole chromosomal aberrations (Figure 6C). Seven metaphases could be examined, and we found that the relative telomere contents in p-arms were lower compared to q-arms for the chromosome pairs 7, 12 and 2, while it was the opposite for the chromosome 1 pair. These findings are in agreement with previous observations by Wise et al. [30]. Twenty metaphases with +X, +Add were also examined with respect to the relative telomere content in the chromosome arms of 7p, 7q, der(7p), der(7q), 12p, 12q, der(12p) and der(12q), and as controls, in 1p, 1q, 2p and 2q (Figure 6C). First, we found that the relative telomere content in all examined chromosome arms was generally lower when compared with chromosome arms in +X metaphases, except for chromosome arms 1q, 2p and 12p, which were equivalent. Secondly, the significant differences in +X metaphases between 7p/7q, 12p/12q, 1p/1q and 2p/2q were eliminated in +X, +Add metaphases. Interestingly, in +X metaphases, the telomere content of 1p was larger than in 1q, whereas it was the opposite in +X, +Add metaphases. Thirdly, the p- and q-arms of the derivative chromosomes der(7) and der(12) had higher telomere contents than their respective q-arms. Together, these findings indicated that the telomere contents of the p-arms on der(7) and der(12) had increased to become bigger than their respective q-arms, especially at 7p, which was involved in the chromothriptic event.

## 4. Discussion

Congenital aneuploidies confer increased rates of certain malignancies [14]. Children with Down’s syndrome (trisomy 21) are up to 15 times more likely to develop leukemia than unaffected children [16,31], and men with KS have an increased risk of breast cancer [32] and germ cell tumors [33,34]. In KS, there are conflicting results as to whether KS involves an increased risk of hematological malignancies. In 1961, the first report about the association of KS with acute myeloid leukemia in a 36-year-old patient was published [35]. In 1963, the first report on KS and acute lymphoblastic leukemia in a 32-year-old patient was then published [36]. A recent cohort study showed that the prevalence of KS is no different in males with pediatric B-ALL than in males in the general population, and that there could even be a protective effect of KS on the development of B-ALL [21].

A search for the karyotype 47,XXY in our local cancercytogenetic registry, which contains cytogenetic information on approximately 28,000 malignant hematology cases, analyzed in January 2001 to December 2019, revealed no previous cases with KS and pre-B ALL. However, we identified four previous cases of KS (or mosaic KS) diagnosed with chronic lymphocytic leukemia, myelodysplastic syndrome, acute myeloid leukemia and chronic myeloid leukemia. This reflects a very low incidence of 0.018% (5/28,000) of KS associated with a hematological malignancy.

A survey of the literature on the co-occurrence of ALL and KS confirmed its rarity [20,37]. The survey included a total of 25 previously reported KS patients with B-ALL since 1963, as listed in Table 2, including our presented case. In 21 cases, results from pretreatment bone marrow cytogenetics were available for evaluation of acquired chromosomal abnormalities in their leukemic cells. The spectrum of acquired chromosomal abnormalities in leukemic cells of KS has not been systematically reviewed previously.

In 71.4% (15/21) of cases, the leukemic cells had acquired additional chromosomal abnormalities, while the constitutional “normal” karyotype for KS patients was observed in 28.6% (6/21) (Figure 7).

Most of the KS cases (93.3%, 14/15) had acquired two or more additional chromosomal aberrations, whereas only 6.7% (1/15) had acquired one additional chromosomal aberration, which was a three-way translocation t(9;22;11). The most frequently observed recurrent chromosomal abnormality was hyperdiploidy with ≥50 chromosomes, which was found in 40% (6/15) of KS cases harboring the additional chromosomal aberrations. Another recurrent chromosomal abnormality in pre-B ALL is translocation t(1;19), which was observed in one case, while other frequent recurrent abnormalities such as t(12;21)(q22;p13), KMT2A rearrangement and iAMP21 were not observed. Interestingly, chromosome 7 abnormalities were observed in 40% (6/15) of the KS cases, of which none were recurrent. The chromosomal findings in our present case have not been reported previously. The high fraction of KS patients (67.7%, 14/21) with two or more acquired chromosomal aberrations in all cytogenetic, evaluable B-ALL KS cases suggests that congenital aneuploid cells are associated with chromosomal instability. This finding is supported by previous studies showing that congenital aneuploid cells, in general, involve increased rates of chromosomal instability [3,4]. It is arduous to investigate chromosomal instability in naturally occurring congenital aneuploidy of a non-cancer origin because most whole-chromosome aneuploidies in humans are fatal except for trisomy 21 (Down’s syndrome), trisomy 18 (Edwards syndrome), trisomy 13 (Patau syndrome) and variation in the copy number of sex chromosomes. In one study, lymphocytes isolated from individuals with trisomies 13, 18 and 21 were treated with phytohemagglutinin (PHA), which is a mitogen stimulating CD3+ cells to proliferate. These were subjected to ploidy analysis by FISH using probes for three different autosomes (chromosomes 8, 15 and 18). For each of these chromosomes, samples from the trisomic individuals had more than double the number of aneuploid cells compared to healthy euploid individuals [3]. In a similar study with lymphocytes from monosomy X (Turner syndrome), the number of aneuploid cells was almost twice that for the control [4]. Although the number of studies is limited, the existing data suggest that cells from congenital aneuploidies may be less chromosomally stable than cells from normal diploid individuals, and that this increased rate of chromosomal instability may explain the observed elevated risks of certain malignancies.

The primary role of telomeres is to maintain chromosomal stability. Telomere dysfunction is linked to chromosomal instability either via progressive telomere shortening or telomere aggregation [49,50]. In our presented three-year-old KS case, we found a three-to-fourfold reduction in telomere length in interphase nuclei harboring acquired additional chromosomal aberrations when compared with constitutional cells containing the inherited extra X chromosome. Interestingly, previous studies have shown that young KS patients (aged 18–24 years) exhibit longer telomeres compared to healthy euploid controls, followed by a steeper slope of telomere attrition, leading to a similar telomere length to that in middle-aged KS individuals [51]. Down’s syndrome (trisomy 21) and Edwards syndrome (trisomy 18) patients also exhibit longer telomeres, which were shown to be present at birth, unlike euploid controls [52]. However, at eight years of age, the telomere length was already shorter than the euploid controls, suggesting more rapid telomere attrition in trisomy 21 cases [53,54]. The telomere attrition rate in childhood KS patients is unknown as there are apparently no published studies on the telomere length in KS patients younger than 18 years of age. This may relate to the fact that most KS patients go undiagnosed due to a lack of clinical abnormalities up to puberty or at older ages. Most studies on telomere lengths and disease processes have been related to telomere shortening. However, recent studies have indicated that there are disease processes caused by both short and long telomere length extremes, indicating that the telomere length in humans has finite upper and lower boundaries [55,56]. A GWAS study of more than 95,000 individuals found that long-telomere-associated SNPs were also associated with an increased risk of cancers, especially melanoma and glioma [57]. It was hypothesized that short telomeres may limit the proliferative potential of mutation-bearing cells, while a longer telomere length may be permissive for increased replicative potential, which in turn, allows the acquisition of additional genetic changes that lead to carcinogenesis [56]. Together, these observations and our findings in KS patients indicate that congenital aneuploid cells with additional chromosomal aberrations exhibit a shorter telomere length, despite how the telomere length might initially have been longer. Whether it is an increased telomere attrition rate in congenital aneuploidies that influences the increased rate of chromosomal instability, and subsequently, the increased risk of malignancy, is presently uncertain and needs further investigation.

The lengths of telomeres vary within a cell not only between the different chromosomes [58,59] but also between individual chromosome arms [30] in euploid cells. In our congenital aneuploid KS case, we examined the telomere lengths of individual chromosomes and chromosome arms involved in the complex rearrangement between chromosomes 7 and 12 and in control chromosomes 1 and 2. First, we confirmed that the average telomere length in his leukemic metaphase is reduced in the same order of magnitude as seen in the interphase nuclei when compared with constitutional cells. Secondly, we found that in constitutional metaphases, the 2p, 7p and 12p telomeres are significantly shorter than their respective 2q, 7q and 12q telomeres, while these differences are minimized in the leukemic metaphases. At 1p, the telomere is longer than at 1q in the constitutional cells, while in the leukemic cells, the 1q telomere is longer than 1p. When examining the telomere length in the leukemic cells of aberrant chromosomes der(7) and der(12), the p-arms are significantly longer than their respective q-arms, indicating an acquired telomere lengthening that is associated with the complex chromosomal rearrangement in these chromosomes. In the control chromosomes 1 and 2, in the constitutional cells, the 1p telomere was longer than the 1q telomere, but the opposite was true for chromosome 2, while the leukemic 1p tended to be shorter than 1q, with no difference for chromosome 2p and 2q. A telomere study of congenital trisomy 21 showed that all p-arm telomeres are longer than those on q-arms, while for congenital trisomy 18, the q-arm telomeres are longer than their corresponding p-arms [52]. There are no published data available on telomere lengths in tumor cells derived from congenital aneuploid cells. The underlying mechanisms for telomere length regulation are presently unclear, although it is well-known that telomere lengthening can be accomplished via two different mechanisms: either via telomerase activation or recombination-mediated telomere pathways referred to as alternative lengthening of telomeres (ALT) [60]. In our presented case, it is plausible that the observed telomere lengthening at specific chromosome sites might have been mediated by an ALT mechanism involving telomere fusions between chromosomes 7 and 12 against a background of generalized cellular telomere shortening. Our finding of a reduced telomere length suggests that telomere dysfunction in young KS patients may be a contributor to pathogenesis.

In our present case, we found evidence of chromothripsis involving the short arm of chromosome 7, which was also involved in an unbalanced translocation with chromosome 12. Chromothripsis is a complex class of structural genomic rearrangements involving the apparent shattering of an individual chromosome or chromosome arm into tens to hundreds of fragments. Chromothripsis has been linked with critically short telomeres as these are recognized as DNA breaks leading to end-to-end fusions due to repair mechanisms, and possible dicentric chromosomes, which cannot be segregated properly at anaphase and result in the formation of break-fusion-bridge cycles [61,62]. Initially, it was thought that chromosthripsis was a rare phenomenon, but further studies have uncovered how is more common than previously believed. Chromothripsis cannot be detected by chromosome banding or FISH analyses; instead, it needs DNA microarray analyses or NGS-based techniques to be applied for it to be uncovered. Therefore, it is not possible to determine whether there are other chromothriptic cases among the KS cases with pre-B-ALL listed in Appendix A.

By aCGH analysis, we found that the SKAP2 (SRC kinase adaptor phosphoprotein 2) gene at 7p15 was highly amplified in the leukemic cells of our KS case, probably because of the chromothriptic event involving the short arm of chromosome 7. SKAP2 is widely expressed in lymphohematopoietic cells and plays a central role in multiple physiological processes, including control of actin-polymerization, response to TGFbeta, integrin signaling and cell migration. A recent study indicated that SKAP2 is involved in the innate immune response and may prevent an excessive inflammation response [63]. How this relates to our present case needs further study. The aCGH analysis further revealed that the MAD1 gene located at 7p22 was deleted. MAD1 is a spindle assembly checkpoint-associated protein, which together with PIGN, is necessary for proper chromosomal alignment and segregation [64,65,66]. The authors showed that suppression or loss of PIGN resulted in MAD1 downregulation and vice versa, accompanied by chromosome instability due to an increased frequency of segregation errors. These observations might explain the observed high hyperdiploidy seen in several pre-B-ALL KS cases but needs further investigation. The TWIST1 gene was also deleted in our KS case. Interestingly, it was recently shown that the TWIST1 gene is a novel regulator of hematopoietic stem cell maintenance through modulation of mitochondrial function [67], but how this finding might correlate with our observed chromosomal instability in KS will need further investigation.

## 5. Conclusions

We characterized a three-year-old boy with B-ALL and the congenital aneuploid syndrome KS carrying the constitutional 47,XXYc karyotype, and performed a literature search where we identified 25 additional KS cases with B-ALL between 1963 and 2016 (Table 2). In 21 of the KS cases, a karyotype at B-ALL diagnosis was available, which showed that in 71% (15/21) of the cases, additional somatic chromosomal aberrations were acquired, and that 93% (14/15) of these cases had acquired two or more additional aberrations, suggesting chromosomal instability and also that congenital aneuploidy alone is not sufficient for leukemogenesis. The most common aberration was high hyperdiploidy, which is also very frequent in non-KS B-ALL, while other frequent abnormalities in B-ALL (e.g., t(12;21)) were not observed in the KS B-ALL cases. It is unfortunately not possible to carry out further cytogenetic analyses of these historic samples to examine for possible factors that might be important for the observed chromosomal instability. We, therefore, took advantage of our newly diagnosed KS in a three-year-old boy with B-ALL to enhance cytogenetic analyses, with the purpose to identify possible chromosomal factors involved in chromosomal instability. The KS was unexpectedly identified during routine cytogenetic diagnostics, and follow-up, in a three-year-old boy with B-ALL. At diagnosis, we identified a complex karyotype, including chromothripsis in his leukemic blasts in addition to his constitutional aneuploid XXY sex chromosomes. Telomere dysfunction is one mechanism that can promote not only the appearance of structural chromosomal aberrations but also the appearance of numerical aberrations that precede and predispose malignant transformation in the human hematopoietic compartment [68]. We, therefore, examined the telomere length by QFISH analysis in interphase nuclei and metaphases. The QFISH method is a single-cell-based method for telomere length analysis that allows for telomere measurements in different cell types according to specific chromosomal aberrations without having to sort cells in advance [26]. Using this method, we found a significant reduction in telomere length in his leukemic blasts when compared with his non-leukemic constitutional aneuploid cells. In metaphases, we found short telomeres on 7p and 12p in the patient’s constitutional cells, suggesting an association with telomere dysfunction and activation of alternative telomere lengthening (ALT pathway) to maintain a minimal telomere length through the selection of fit clones, which bypass the senescence checkpoints, thereby promoting tumorigenesis and sustaining cell survival.

Together, the presented data, including the literature review, indicate that congenital aneuploidy in B-ALL might be associated with chromosomal instability, which may relate to dysfunctional telomeres via progressive telomere shortening or telomere aggregation. Additional studies are, however, required to establish such a possible link.

## Figures and Tables

**Figure 1 cancers-14-02316-f001:**
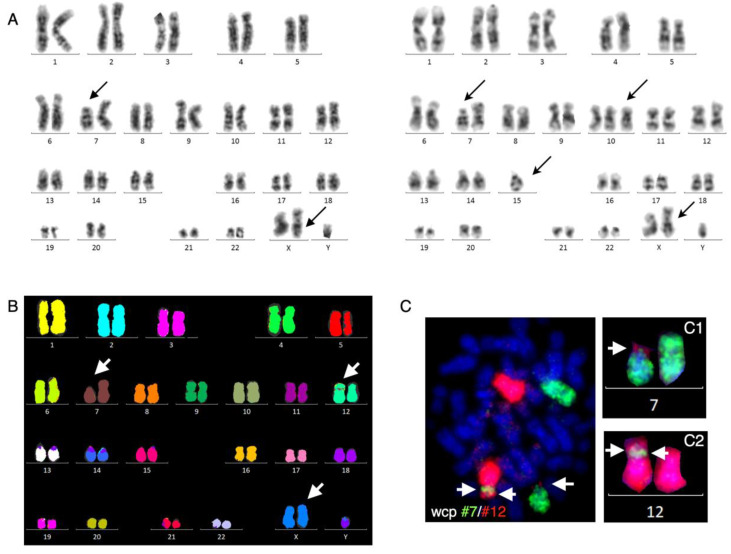
G-banding, 24-color karyotyping and dual-color whole-chromosome painting. (**A**) G-banded karyotype at diagnosis showing two hyperdiploid clones with 47 chromosomes due to an additional X chromosome, in which one sub-clone had an apparent del(7)(p13) (left panel) and the other sub-clone—in addition to the apparent del(7)(p13)—had an additional chromosome 10 and loss of chromosome 15 (right panel). (**B**) At diagnosis, 24-color karyotyping confirms the additional X chromosome but also reveals that the sub-clone with an apparent single structural aberration, del(7)(p13), is more complex, with a small insertion of chromosome 7 material on to the short arm of chromosome 12 close to the centromere. (**C**) Dual-color whole-chromosome painting with whole chromosome 7 (green) and whole chromosome 12 (red) probes revealed a minor/cryptic translocation of chromosome 12 material to the aberrant chromosome 7, del(7)(p13) (C1), and two minor insertions of chromosome 7 material on the short arm of chromosome 12 (C2). Aberrant chromosomes are marked by black arrows in G-banding (**A**), or white arrows in 24-color karyotyping (**B**) and a whole-chromosome painting assay (**C**).

**Figure 2 cancers-14-02316-f002:**
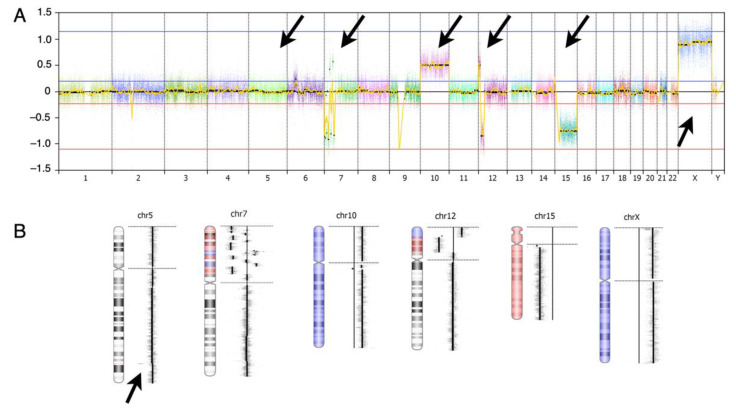
High-resolution 180K aCGH analysis revealed several additional cryptic chromosomal aberrations. (**A**) Whole-genome view depicts chromosomal aberrations at the following regions as indicated by black arrows: deletion at 5q33.3, chromothripsis at 7p12.1p22.2, gain of whole chromosome 10, telomeric gain of 12p13.2p33.33 and centromeric loss of 12p11.23p13.2, loss of whole chromosome 15 and gain of whole chromosome X. Vertical blue lines indicate log2 ratios +0.24 and +0.60 and red vertical lines indicate log2 ratios –0.24 and –1.0. The X-axis at the bottom indicates the chromosomal position. (**B**) Chromosomal view of individually affected chromosomes. To the left of each chromosomal array profile are respective ideograms representing each chromosome, in which the red and blue overlay indicate the loss and gain of chromosomal material, respectively. Aberrant chromosomes are marked by black arrows.

**Figure 3 cancers-14-02316-f003:**
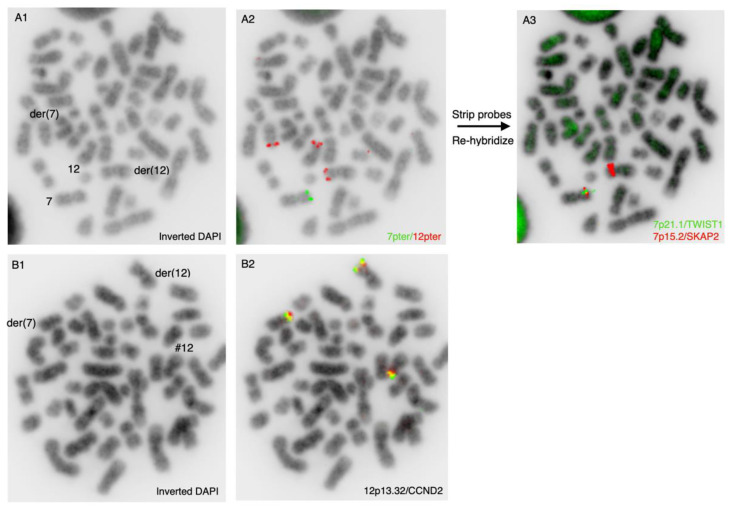
Dual-color locus-specific FISH analyses on metaphase spreads. Upper row represents hybridization using subtelomeric probes from 7pter (green) and 12pter (red) with inverted DAPI (**A**1) and combined color channels (**A**2). After imaging, these probes were stripped off and the microscopy slide was rehybridized with probes representing the TWIST1 gene at 7p21.1 (green) and SKAP2 gene at 7p15.2 (red), depicted as a combined inverted DAPI and color channels (**A**3). Chromosomes 7, 12 and their derivative counterparts are labeled in the inverted DAPI image. Lower row represents hybridization with a dual-color probe at the CCND2 gene at 12p13.32, with inverted DAPI (**B**1) and combined color channels (**B**2). Derivative chromosomes 7 and 12 are labeled in the inverted DAPI image. Inverted grayscale imaging allows for identification of individual chromosomes by banding morphology.

**Figure 4 cancers-14-02316-f004:**
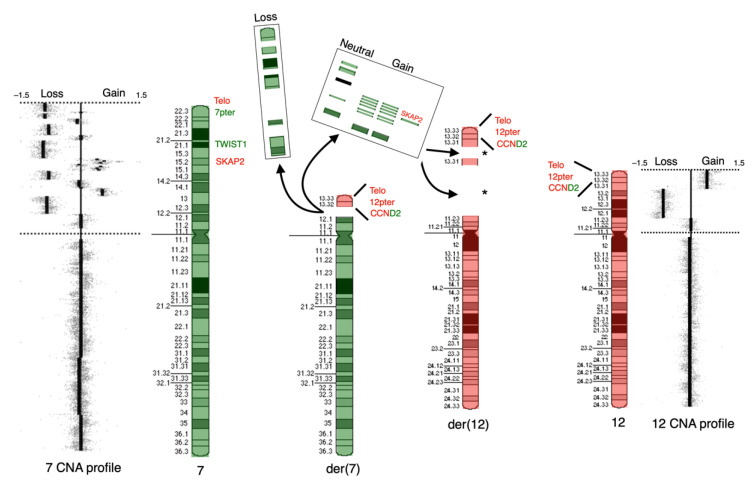
Schematic representation of the complex chromosomal events between chromosomes 7 and 12. aCGH profiles of chromosomes 7 and 12 are shown to the left and right, respectively. Ideograms of normal chromosomes 7 and 12 are presented with green and red overlays, respectively, as well as their derivative counterparts, der(7) and der(12), including breakpoints and the locations of specific regions and genes (telomeres, 7pter, 12pter, CCND2, TWIST1 and SKAP2). The coloring corresponds to the previous FISH experiment in Figure 1. The regions located at the short arm of chromosome 7 that are lost in the complex rearrangement are indicated in the box labeled “Loss,” while the translocated and amplified regions are indicated in the box labeled “Neutral” (copy number neutral) and “Gain”. The asterisks (*) at der(12) indicate the plausible regions for the translocated regions from 7p. It is not possible to show the individual order of final joining of chromosomal fragments given the presented available data.

**Figure 5 cancers-14-02316-f005:**
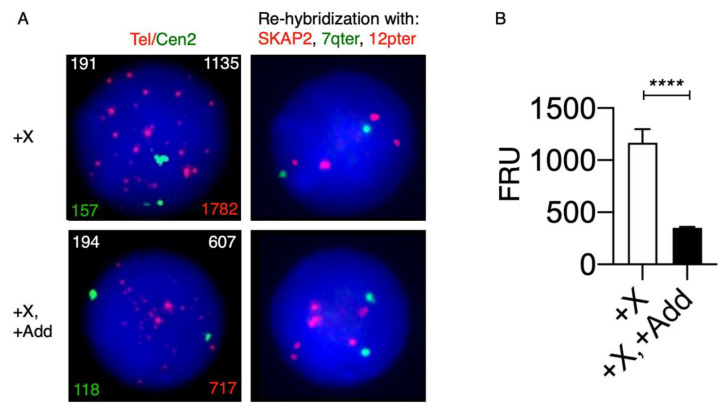
Telomere quantification in nuclei with telomere/centromere fluorescence in situ hybridization (T/C FISH) and re-hybridization with locus-specific probes. (**A**) A microscopy slide was prepared for hybridization with a pan-telomeric (G_3_TA_2_) Cy3-labeled PNA probe (Tel, red), and as the internal control, a FITC-labeled PNA probe for the centromere of chromosome 2 (Cen2, green). After hybridization and washing steps, nuclei were counterstained with DAPI/antifade solution before slide scanning, intensity measurements and quantification using the automated Metafer4 system, as described in the Materials and Methods. More than 200 nuclei were captured automatically for evaluation. After automated scanning and image capture, the slide was removed from the scanning system and Tel/Cen2 PNA probes were stripped before re-hybridization with locus-specific probes (SKAP2 (red), 7qter (green) and 12pter (red)), to discriminate nuclei without additional chromosomal aberrations besides an extra X chromosome +X from nuclei with additional chromosomal aberrations besides an extra X chromosome +X, +Add. A unique feature of the Metafer4 automated scanning system is its ability to relocate the previously captured nuclei for individual evaluation of the hybridization pattern after the locus-specific hybridization. Representative nuclei after automated scanning are depicted, showing a nucleus without additional chromosomal aberrations besides an extra X chromosome +X (upper left-hand panel) and after re-hybridization/re-location of its respective hybridization pattern (4R2G) (upper right panel), and similarly, a nucleus with additional chromosomal aberrations besides an extra X chromosome +X, +Add (lower left-hand panel) and its respective hybridization pattern (≥6R2G) (lower right pattern). In the left-hand panels, the fluorescence intensities of chromosome 2 centromeric (green) and pan-telomeric (red) probes are displayed in the left and right bottom corners, respectively, the ratio between telomeric and centromere 2 fluorescence intensities multiplied by 100 is displayed in the upper right corner and the cell number is displayed in the upper left corner (lefthand panels). (**B**) Mean telomere content defined as FRUs (fluorescence ratio units) after scoring cells with only +X and +X, +Add. Significance testing showed that the telomere content is significantly (*p* < 0.001, ****) lower in cells with additional chromosomal aberrations +X, +Add when compared with cells harboring only an extra X chromosome +X.

**Figure 6 cancers-14-02316-f006:**
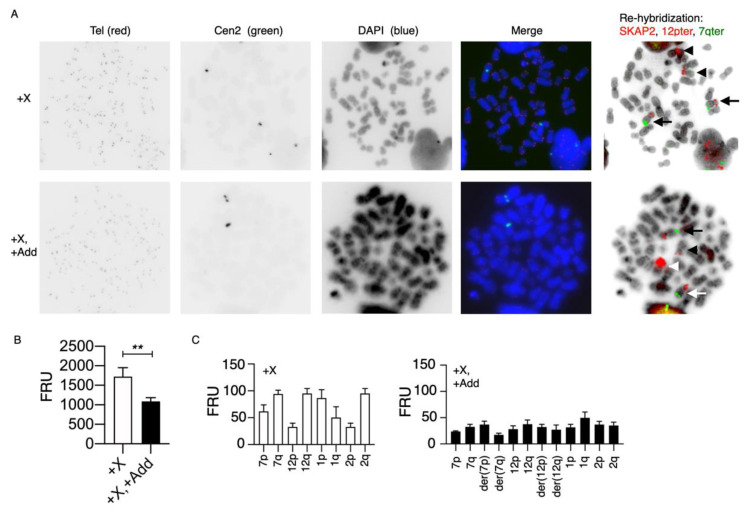
Telomere quantification in metaphases and selected chromosome arms with telomere/centromere fluorescence in situ hybridization (T/C FISH) and re-hybridization with locus-specific probes. A microscopy slide was prepared for hybridization with a pan-telomeric (G_3_TA_2_) Cy3-labeled PNA probe (Tel, red), and as the internal control, a FITC-labeled PNA probe for the centromere of chromosome 2 (Cen2, green). After hybridization and washing steps, metaphases were counterstained with DAPI/antifade solution before slide scanning and automated image capture. More than 30 metaphases were captured. After automated scanning and image capture, the slide was removed from the scanning system and Tel/Cen2 PNA probes were stripped before re-hybridization with locus-specific probes (SKAP2, 7qter and 12pter). Each captured metaphase could then be relocated with the Metafer4 scanning system to discriminate metaphases with different chromosomal aberrations. (**A**) Inverted grey-scale images of representative metaphases in each color channel are depicted together with a merged color image of these channels and merged inverted DAPI image with locus-specific probes after re-hybridization/re-location without additional chromosomal aberrations besides an extra X chromosome +X (upper row) from metaphases with additional chromosomal aberrations besides an extra X chromosome +X, +Add (lower row). The white arrows mark normal chromosome 7 (SKAP2 in red and 7qter in green), the white arrowheads mark normal chromosome 12 (12pter in red), the orange arrow marks der(7) (12pter in red and 7qter in green) and the orange arrowhead marks der(12) (amplified SKAP2 in red). (**B**) Mean telomere content defined as FRUs (fluorescence ratio units) after scoring metaphases with only +X and +X, +Add. The total fluorescence intensity of pan-telomeric (red) and chromosome 2 centromeric (green) probes was measured in each metaphase with Image J software, as described in the Materials and Methods. A significantly lower telomere content in metaphases with an extra X chromosome and additional aberrations, when compared with an isolated extra X chromosome, was confirmed (*p* < 0.01, **). (**C**) Mean telomere content defined as FRUs (fluorescence ratio units) after identification of chromosomes 7p, 7q, 12p and 12q, which are involved in the additional chromosomal rearrangements, and in normal chromosomes 1p, 1q, 2p and 2q, in metaphases scored with only +X (left-hand panel) and in metaphases with +X, +Add, der(7p) and der(12p) as well (right-hand panel). The fluorescence intensity of the pan-telomeric (red) probe was normalized to the chromosome 2 centromeric (green) intensity in each metaphase to measure each chromosome arm.

**Figure 7 cancers-14-02316-f007:**
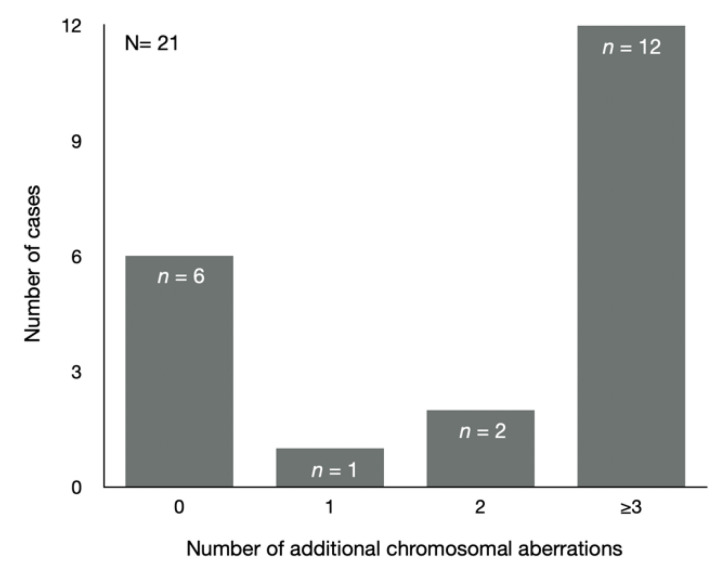
Distribution of number of additional chromosomal aberrations in KS patient cases with B-ALL.

**Table 1 cancers-14-02316-t001:** Copy number aberrations detected by aCGH analysis.

Cytoband	Genomic Position	Copy Number Aberration	Size (Mb)	Selected Affected Genes
5q33.3	157,949,420–158,532,620	loss	0.58	EBF1
7p22.3p22.2	1–4,151,047	loss	4.15	MAD1
7p22.1	4,696,477–7,185,788	loss	2.49	
7p21.3p21.2	9,732,340–14,873,380	loss	5.14	
7p21.1p15.2	17,205,731–25,570,535	loss	8.36	TWIST1
7p15.2	25,727,833–26,358,379	gain	0.63	
7p15.2	26,433,410–26,884,359	gain	0.45	SKAP2 ex. 6–13
7p15.2	26,884,359–26,935,498	high gain	0.05	SKAP2 ex. 1–5
7p15.1	26,935,498–28,097,743	gain	1.16	
7p14.3	29,077,402–29,334,815	gain	0.26	
7p14.3	29,485,473–30,285,280	gain	0.80	
7p14.3	30,445,107–30,871,460	gain	0.43	
7p14.3p14.2	31,012,122–35,289,392	loss	4.28	
7p14.1	38,309,858–38,385,111	homozygous loss	0.08	TARP
7p14.1	38,982,260–42,893,551	gain	3.91	
7p14.1p12.1	42,893,551–50,808,131	loss	7.91	IKZF1
10p15.3q26.3	whole chromosome	gain	135.53	
12p33.33p13.2	1–11,278,401	gain	11.28	CCND2
12p13.2p11.23	11,278,402–27,743,563	loss	16.47	ETV6
15p13q26.3	whole chromosome	gain	102.5	
Xp22.33q28	whole chromosome	gain	155.3	

**Table 2 cancers-14-02316-t002:** Case reports of acute lymphoblastic leukemia associated with Klinefelter syndrome.

Year	Age	Diagnosis	PB ^1^	Diagnostic BM ^2^: Conventional Cytogenetics	Reference
1963	32 yrs	ALL ^3^	47,XXYc	not done	[36]
1966	n.a. ^4^	ALL	47,XXYc	not done	[38]
1974	19 yrs	ALL	48,XXXYc	not done	[39]
1984	9 days	ALL-L2	47,XXYc	not done	[40]
1990	n.a.	ALL	n.a.	47,XXYc,del(7)(q22),add(19)(p?),-20,+mar[19]/47,XXYc[2]	[41]
1992	21 months	ALL	47,XXYc	47,XXYc	[42]
-	3.5 yrs	ALL	46,XY/47XXYc	46,XY/47,XXYc	do
1994	4 yrs	ALL-L2	47,XXYc	47,XXYc	[43]
1999	n.a.	ALL	n.a.	46,XXYc,del(7)(q22),add(19)(p13),-20[19]/47,XXYc[2]	[44]
-	49 yrs	ALL-L2	46,XY/47,XXYc	47,XXYc,t(9;22;11)(q34;q11;q13)[10]/46,XY[6]/47,XXYc[4]	[45]
2002	2.5 yrs	ALL-L1	47,XXYc	54,XXYc,+4,+8,+9,+12,+17,+18,+21[3]/47,XXYc[16]	[20]
2004	17 yrs	B-ALL	47,XXYc[1]/48,XXXY[9]	47,XXYc[5]/46,XY[8]	[46]
2008	3.9 yrs	B-ALL	n.a.	54,XXYc,+4,+8,+9,+14,+16,+18,+21[cp6]/46,XY[28]/47,XXYc [18]	[47]
2016	14 yrs	B-ALL	n.a.	47,XXYc,dic(7;16)(p11;p13),+21[7]	[48]
2016	n.a.	B-ALL	n.a.	47,XXYc[21]	[21]
-	n.a.	B-ALL	n.a.	47,XXYc[20]	do
-	n.a.	B-ALL	n.a.	47,XY,del(X)(q24),del(6)(q13q21),-19,+mar[3]/47,XXYc	do
-	n.a.	B-ALL	n.a.	47,XXYc,der(11)t(11;12)(q13;p13),der(12)t(11;12)(q23;p13) ins(12;11)(q24;q13q23),der(15)t(8;15)(q13;q13)[7]/47,XXYc [4]	do
-	n.a.	B-ALL	n.a.	61,XXYc,+X,+4,+4,+6,del(6)(q21),+10,+11,+14,+17,+18,+19, +20,+21, +21,+mar[cp11]/47,XXYc[9]	do
-	n.a.	B-ALL	47,XXYc	47,XXYc,t(1;19)(q23;p13.3)[5]/47,XXYc,der(19)t(1;19)(q23; p13.3)[6]/ 47,XXYc[21]	do
-	n.a.	B-ALL	47,XXYc	46,XY,-X,inv(4)(p15.2p16),del(9)(p13p12)[12]/45,idem, -7[7]/ 47,XXYc[21]	do
-	n.a.	B-ALL	47,XXYc	58,XXYc,+4,+6,+8,+10,+11,+12,+14,+18,+19,add(19)(p13.3), +21,+21 [cp13]/47,XXYc[13]	do
-	n.a.	B-ALL	47,XXYc	50,XXYc,+X,+20,+21[13]/47,XXYc[7]	do
-	n.a.	B-ALL	n.a.	55,XXYc,+X,+X,dup(1)(q12q32),+4,+6,i(7)(q10),+11,+18,+21,+21[14]/47,XXYc[6]	do
-	n.a.	B-ALL	47,XXYc	not done	do
2022	3 yrs	pre-B ALL	47,XXYc	47,XXYc,del(7)(p13)[2]/47,idem,+10,-15[4]/47,XXYc[19]	present case

^1^ PB: peripheral blood; ^2^ BM: bone marrow; ^3^ ALL: acute lymphoblastic leukemia; ^4^ n.a.: not available.

## Data Availability

The data presented in this study are available in this article (and Appendix A).

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
