# Peer review of "Congenital Aneuploidy in Klinefelter Syndrome with B-Cell Acute Lymphoblastic Leukemia Might Be Associated with Chromosomal Instability and Reduced Telomere Length"

_cancers, 2022, doi:10.3390/cancers14092316_

Round 1

Reviewer 1 Report

The paper by Kjeldsen reports a case report of a three years old child with KS and an additional B-ALL, which leads to additional chromosomal aberrations, suggesting that the congenital aneuploidy linked to B-ALL is associated with chromosomal instability triggered by the reduction of telomere length. At the same time, a review of the previously described KS cases with B-ALL is reported.

Major concern:

My primary concerns are not about the paper itself, that it seems complete to me and it scientifically sounds but on author's lists.

I found that the case report is complete and well-presented and the patient is well characterized, both genetically and clinically. Moreover, the cytogenetic study is very detailed (in some passages even too much), and I think it requires an excellent technical background for conducting the wet laboratory experiments and reporting. Thus I believe that "the biotechnologist Pia Kristensen who gave the excellent technical assistance" should be listed in the author list instead of being only cited in the acknowledgment section. The same is true for the clinicians who referred the patient and shared the information.

Another point:

 the author concludes that congenital aneuploidy in B-ALL is associated with chromosomal instability, admitting that further studies are needed to demonstrate this hypothesis. I would use a more hypothetical tone along with the text, since he does not have all the data for demonstrating this idea (although the hypothesis is undoubtedly valuable). For example, I found that only 21 cases are not enough to come to such a conclusion. I should change the title accordingly.

I would add the literature research method used for case retrieval to the method list.

Minor:

The title contains a typo error: 'Lymphblastic’ instead of lymphoblastic

Do not relegate the table with the list of previously described cases in the supplementary material. I found it very helpful in the main body of the manuscript.

Fig.7: complete the y axis title…numbers of what?

Author Response

Major points of Reviewer 1

Comments and actions taken by the author

Point 1:

My primary concerns are not about the paper itself, that seems complete to me and it scientifically sounds but on author’s list.

I found that the case report is complete and well-presented and the patient is well characterized, both genetically and

clinically. Moreover, the cytogenetic study is very detailed (in some passages even too much), and I think it requires an excellent technical background for conducting the wet

laboratory experiments and reporting. Thus, I believe that "the biotechnologist Pia Kristensen who gave the excellent

technical assistance" should be listed in the author list instead of being only cited in the acknowledgment section. The same is true for the clinicians who referred the patient

and shared the information.

I would like to thank the reviewer for bringing this important subject into perspective.

From the international research environment there are several specific views on who should be an author or a contributor when data are published. However, it is generally agreed that everyone who is listed as an author should have made a substantial, direct, intellectual contribution to the work. For example (in the case of a research report) they should have contributed to the conception, design, analysis and/or interpretation of data. Honorary or guest authorship is not acceptable.

More specifically, the International Committee of Medical Journal Editors (ICMJE) developed 4 criteria for authorship that can be used to distinguish authors from other contributors. The criteria are: Substantial contributions to the conception or design of the work; or the acquisition, analysis, or interpretation of data for the work; AND Drafting of the work or revising it critically for important intellectual content; AND Final approval of the version to be published; AND Agreement to be accountable for all aspects of the work in ensuring that questions related to the accuracy or integrity of any part of the work are appropriately investigated and resolved. Contributors who meet fewer than all 4 of the above criteria for authorship should not be listed as authors, but they should be acknowledged.

It is greatly appreciated that the reviewer acknowledges the work of my highly skilled technician (I know I previously entitled her biotechnologist but technician is more correct or appropriate). She indeed is a highly skilled technician and invaluable in our routine cytogenetics work. In the present work she has been involved in acquisitions of data but she has not been involved in conception, design, analysis or interpretation of the work.

The clinicians endorsed the study but they felt that it was too specialized for them to contribute substantially in such a highly specific cytogenetic study.

For the reasons given above I have with great consideration re-evaluated the author’s list which has not been altered.

Point 2:

The author concludes that congenital aneuploidy in B-ALL is associated with chromosomal instability, admitting that

further studies are needed to demonstrate this hypothesis. I would use a more hypothetical tone along with the text, since he does not have all the data for demonstrating this

idea (although the hypothesis is undoubtedly valuable). For example, I found that only 21 cases are not enough to come to such a conclusion. I should change the title

accordingly.

The title has been changed so it uses a more hypothetical tone. Similar changes have also been applied at relevant places such as “Simple summary”, “Abstract” and ”Conclusions”.

Point 3:

I would add the literature research method used for case retrieval to the method list.

The method for retrieving the literature cases have been included as a separate paragraph in the Materials and Methods section.

Minor points of Reviewer 1

Comments and actions taken by the author

Point 1.

Title contain a typo error: “Lymphblatic”

“Lymphblatic” has been corrected in title

Point 2.

Do not relegate the table with the list of previously described cases in supplementary material. I found it very helpful in the main body of the manuscript.

The requested change was made and is now included in the body of the text.

Point 3.

Fig. 7.: complete the y-axis title … numbers of what?

The requested completion in the figure has been made

Reviewer 2 Report

Kjeldsen presents a well characterized case report of 3 years old boy with B-ALL and Klinefelter syndrome.

However the study is observational as the conclusion of enhanced chromosomal instabilities occurrence in similar patients obtained by literature mining lacks adequate methodologies (sample size, exclusion inclusion criteria, quality assessment) and controls (comparison with chromosomal instability in B-ALL and other lymphomas in absence of Klinefelter syndrome).

Thus novelty/interest is low and conclusions not supported by strong evidences.

Author Response

Comments by Reviewer 2

Actions taken by Authors

Kjeldsen presents a well characterized report of a 3 years old boy with B-ALL and Klinefelter syndrome.

However, novelty/interest is low and conclusions not supported by strong evidences

None taken.

Round 2

Reviewer 2 Report

No substantial change made apart the insertion of table 2, the article remain a well done case report.